# Developing and Implementing a Narration of Care Framework to Teach Nurses When and How to Narrate Care

**DOI:** 10.3390/nursrep15070244

**Published:** 2025-07-02

**Authors:** Courtenay R. Bruce, Natalie N. Zuniga-Georgy, Nathan Way, Lenis Sosa, Emmanuel Javaluyas, Terrell L. Williams, Gail Vozzella

**Affiliations:** 1System Patient Experience, Houston Methodist System, Houston, TX 77030, USA; nzuniga@houstonmethodist.org (N.N.Z.-G.); tlwilliams@houstonmethodist.org (T.L.W.); 2Department of Nursing, Houston Methodist the Woodlands Hospital, The Woodlands, TX 77385, USA; nmway@houstonmethodist.org; 3Center for Nursing Research, Education and Practice, Houston Methodist Hospital, Houston, TX 77030, USA; lsosa@houstonmethodist.org; 4Department of Nursing, Houston Methodist Hospital, Houston, TX 77030, USA; ejavaluyas@houstonmethodist.org (E.J.); gmvozzella@houstonmethodist.org (G.V.)

**Keywords:** nursing communication, nursing education, nursing framework, nurse–patient relationship, patient care assistant

## Abstract

**Background:** It is generally well-known that narration of care is critically important to high-quality nursing care. Narration of care is loosely defined as a nurse’s ability to describe to patients and families the clinical purpose behind nursing practice, what is hoped to be achieved, and the “why” (or clinical rationale) behind nursing activities. Despite the importance of narration of care, there is little practical guidance given to nurses about how to narrate care—what makes for effective or ineffective narration of care. **Objective:** Our aim was to develop a framework for teaching nurses and patient care assistants (PCAs) on how to effectively narrate care. In this article, we provide a practical framework for teaching nurses and PCAs how to narrate care. We describe the process of developing the framework as part of quality improvement efforts and implementing a course for eight hospitals based on the framework. **Methods:** Consistent with a Plan-Do-Study Act (PDSA) quality improvement approach, we developed the framework by first conducting a data and literature review, then convening a taskforce, discussing with patients on our existing committees, and finally formulating a framework. We then drafted supplementary cases and course material and implemented a course to teach nurses and PCAs how to narrate care. **Results:** The narration of care framework (NOC) that we developed and implemented consisted of the following five principles, which can be called RECAP as an acronym: 1. The “R” in RECAP stands for removing uncertainty. 2. The “E” in RECAP stands for explaining the environment. 3. The “C” in RECAP stands for being calm and sincere. 4. The “A” in RECAP stands for assume nothing. 5. The “P” in RECAP stands for personal connection. As for the course developed based on the RECAP principles, there was a total of 276 course offerings conducted by 30 facilitators, and 7341 nurses and PCAs completed the course. The evaluations reflected that 99% of learners believed their learning was improved by the course. **Discussion:** There are several multifaceted benefits to NOC: nurses’ and PCAs’ capability to narrate care well shows empathy and compassion to patients; it strengthens patient understanding and education that can lead to improved patient outcomes; and it helps allay patients’ uncertainties and anxieties. In essence, narrating care in an effective manner cultivates a strong nurse–patient therapeutic relationship. Yet, in the absence of any practical guidance, nurses and PCAs are left to develop narration skills on their own, learning by trial and error, and, in doing so, perhaps failing to meet patients’ needs and failing to fully derive the many benefits that the NOC is designed to achieve. Our hope is that, if hospital systems adopt our work, nurses and PCAs can comfortably and confidently enter the profession knowing the purpose or narrating care, its many benefits, and how to practically conduct sufficient narration, and what would constitute insufficient narration. Hospitals, in turn, can specify and clearly articulate their expectations for nurses and PCAs narrating with patients—what would make for a strong, compassionate process and what would be inadequate. For more experienced nurses, they can use the RECAP framework to reflect on their own practices and perhaps strengthen or refreshen existing skills. **Conclusions:** NOC is acknowledged, somewhat implicitly, as being critical to nursing and PCA practice, yet practical instruction and specified principles are lacking. We aimed to fill this gap by developing, implementing, and teaching a practical framework, armed with many tools nurses can use.

## 1. Introduction

There is general agreement that developing and sustaining a caring nurse–patient relationship is essential to delivering high-quality nursing care [1]. To cultivate a therapeutic nurse–patient relationship, the nurse must possess and demonstrate appropriate caring behaviors, including a caring attitude, empathy, and strong communication (verbal, non-verbal, and listening) skills [2].

There are three stages of developing and maintaining a nurse–patient relationship [3]. First, during the orientation phase, the nurse establishes a relationship and sets the tone for future interactions by eliciting information via assessment, history-gathering, and patient interviewing. Second, during the working phase, the nurse offers information and education to promote patient learning, enhance recovery, and strengthen well-being. The third and final stage involves terminating the relationship by summarizing care information and providing work closure between the nurse and the patient [4].

It is the second phase—the working phase—that is likely the most substantive component of the nurse–patient relationship, and yet, it is the phase that is most underdeveloped. There is research available on the behavioral aspects (what the nurse carries out) during the working phase, and measures exist to evaluate behavioral aspects, attitude, and affect [4]. A significant gap in the literature exists in that, to our knowledge, little has been written on the communication aspect (what the nurse says) during the working phase to promote and maintain a healthy and therapeutic relationship.

There seems to be an implicit understanding that narrating care is important for patient education during the working phase. A recent Google search for “nursing patient education” yielded over 566,000 results. Yet, it is unclear how nursing and patient care assistant (PCA) staff should learn what to say to patients during the working phase. In other words, without explicit guidance on how to narrate care, the nursing and PCA staff are prone to narrating in a way that is substandard, thereby undermining the working phase of the nurse–patient therapeutic relationship.

To fill this gap, our aim was to develop a framework for teaching nurses and PCAs on how to narrate care. In this article, we describe the process of developing the framework as part of quality improvement efforts and implementing a course for eight hospitals based on the framework. Our quality improvement goal was to teach nurses and PCAs how to narrate care, i.e., what to say and when to say it, during the working phase, to promote and maintain the nurse–patient relationship.

There is, to our knowledge, no established definition of what it means to narrate care. Therefore, we define it as the nursing and PCA staff’s ability to describe to patients and families the clinical purpose behind nursing and PCA practice, what is hoped to be achieved, and the rationale for staff tasks and behaviors, a definition we formulated through the task force efforts discussed in the Section 2 below.

## 2. Methods

### 2.1. Developing a Framework

Conducting a Data Review and Literature Review

Plan-Do-Study-Act (PDSA) is a quality improvement process that allows organizations to implement change in four phases: First, planning to test a change (plan), carrying out a project or process change (do), observing and analyzing results of the project through measured tools (study), and then deciding whether to scale-up the initiative as is, sun-set it as is, or make modifications and refinements before scaling [5].

Consistent with robust data gathering that would be needed as part of the planning phases, we first assessed whether and to what extent there was a narration of care deficiency in our hospitals. The Hospital Consumer Assessment of Healthcare Providers and Systems survey (or HCAHPS) is a national, government-mandated, public-reported survey of patients’ perspectives of hospital care [6]. In reviewing our HCAHPS data across seven hospitals in July 2023, we noticed that “nurse explanation and information” and “nurse addressed concerns” were flagged as high correlation coefficients impacting overall scores. Specifically, on the questions, “I understood how to manage my care”, and “understood purposes and side effects of medication”, and “ease of finding someone to talk to in order to address concerns”, were all flagged as having correlation coefficients above 0.55, meaning that they were all correlated with “overall rating of care”, and yet all falling outside of the top 25% of national percentile ranks.

This was our first indication that there was likely a deficiency in how our nursing and PCA staff were narrating care during the working phase and that extensive quality improvement efforts were needed. The second indication was found in a thematic analysis we conducted of patient comments in HCAHPS surveys between 2022 and 2023, which elucidated that patients felt “anxious”, “uncertain”, and “did not know what to expect”, suggesting that healthcare professionals were missing opportunities to identify patients’ non-verbal and verbal cues indicating uncertainty and alleviating them through the effective NOC during the working phase. Thematic analysis is a method wherein coders elicit key themes that emerge from statements, using an inductive approach, to allow for new themes to emerge without an existing framework [7].

2.Convening a Task Force

Recognizing that there was likely a systemic issue requiring a quality improvement intervention at our hospitals, we continued the planning efforts of the PDSA model by convening a task force of 25 nurse leaders, patient experience specialists, and education specialists (called as the “NOC Committee”) from across the system to define and articulate what it means to narrate well. To build the Committee, we elicited (and purposively selected) the involvement of acute care nurse leaders from each hospital, and we asked them to identify other members of their team who were strong narrators who would have an interest participating in the project. The patient experience team consisted of 1–2 people already hired into a patient experience position who served in that role for their hospital. By selecting individuals from each hospital, we could ensure a multidisciplinary, multi-institutional Committee.

3.Conducting Patient Discussions

The second task of the NOC Committee was to meet with patients. To ground this work, we met with our patient and family advisory councils that already preexist as part of our hospitals’ quality improvement efforts, who we meet with routinely to reflect on their hospital experiences and to guide where and how we need to improve.

During these Council meetings, along with our daily rounding practices, we were able to purposively sample 50 diverse, English-speaking patients, with a range of medical conditions and surgical procedures, who had previously indicated they were anxious, uncertain, or felt a lack of preparedness. We handwrote notes to avoid sensitivities introduced by having a tape-recorder in a patient room and destroyed these notes after compiling the data into Microsoft Excel; coding was then performed using thematic analysis techniques, a process described above.

Several researchers have noted the important role that patient education and therefore, by extension, likely NOC, can have on reducing patient anxieties [8,9], and the frequency of patient comments suggested that narration was substandard.

Consistent with other researchers’ findings, in our patient discussions, we found that several patients felt that nursing and PCA staff were primarily focused on tasks and made assumptions about patients’ needs and concerns and, in doing so, missed opportunities to narrate care in a way that would reduce anxieties and uncertainties [10].

Task-driven behaviors have been a critique lately in the nursing literature [11], with several researchers contending that, as administrative nursing tasks and patient-volume workloads have increased, nurses have had to find shortcuts to complete their work [12,13]. In doing so, they may shorten their nurse–patient interactions, explanations, or narrations in order to complete their mandatory clinical or administrative tasks.

There is some support for this finding in that many of our patients relayed that the listening component of communication was sometimes missing in their nurse–patient interactions. Specifically, attending behavior, defined as a physical demonstration of nurses’ readiness and willingness to listen was, in the minds of many of our patients, a critical element that was missing in some nurse–patient interactions. Consistent with findings from other researchers, our patients noted that many nurses were so task-focused due to the complexity of care that non-verbal behaviors signaling approachability and a genuine interest in understanding the patient were occasionally overlooked or insufficiently demonstrated.

Thus, the patient discussions and our in-depth review of HCAHPS scores and comments informed the development of our framework in that we knew three things:4.Unilateral communication, in which clinical staff speak *at* or *to*, rather than *with* patients, can give the impression that staff are primarily task-oriented rather than person-centered. To address this, NOC must extend beyond one-way patient education and incorporate elements of presence and connection, emphasizing relational engagement over transactional interaction.5.NOC would need to include an “uncertainty” component. Anxiety, lack of preparedness, and uncertainty were prominent themes tied to HCAHPS nursing communication scores, particularly questions about “education” and “information”, which suggested that patients viewed nursing staff as being integral to relieving their anxieties and uncertainties through the use of patient education and information, or narration.6.NOC would need to include some component of listening. Since our patients noted a perceived lack of attentive behaviors, we knew that non-verbal and verbal behaviors suggesting approachability would need to be incorporated.7.Formulating a Consensus Framework

Armed with the findings from our HCAHPS data, patient survey comments, patient discussions, and literature review, our NOC Committee met several times to iteratively develop our NOC and cases, thereby undertaking the “Do” component of the PDSA quality improvement framework.

As described above, we knew that removing uncertainty and alleviating patient anxieties would be foundational to the framework, as well as using non-verbal and verbal behaviors to demonstrate approachability.

Thus, at this point in our work, we were able to develop 3 elements of the NOC framework:8.*Remove Uncertainty:* By explaining what we are performing, what the patient can anticipate, and why it is important for their care.9.*Calm and Sincere:* Leave your own negative emotions at the door, including conveying a sense that you are rushed. When speaking with patients, convey calmness in your tone and sincerity with your posture by being open, facing them, moving close, and making eye contact.10.*Personal Connection*: Make a personal connection by making notes of what is important to your patient to reference later and listen intently to matters that concern them. Updating the patient communication board with what they share is most important for them for the day (rest, contacting a family member, etc.). When patients feel cared about, they share concerns and feelings—details that can lead to safer care and better outcomes.

To further refine our concept, we met with two system-shared governance councils (the Magnet Council and the System Acute Care Council), which consisted of 30 bedside nurses, PCAs, and nurse leaders, to learn more from nurses on high-performing HCAHPS on *how* they would remove uncertainty. Many of them noted that they explicitly refer to equipment, safety measures, or other unique hospital environmental factors that could create uncertainty for their patients to explain the purpose behind them. By explaining the “why” and “how” behind the environment, they felt that their patients were more assured.

Other nursing and PCA staff noted that they make no assumptions about their patients. Regardless of the patients’ experiences, the nurses start with foundational concepts to explain why they were performing a particular task and what was hoped to be achieved. By assuming nothing, they felt their patients were more at ease with their narrations.

Based on the feedback from these two councils, we iteratively revised the NOC framework to include two additional components:11.*Explain the Environment:* By providing the purpose behind equipment, monitor readings, and alarms, and reassuring patients of their safety if an alarm is triggered or readings vary.12.*Assume Nothing:* Just because your patient has been in the hospital before, is a healthcare worker, has higher education, or has chronic conditions, do not assume they are confident and proficient in their state of being or care. Always give clear verbal information to describe what is happening, what to expect, and why it is occurring, so that the patient is able to understand.

Thus, in summary, the NOC framework we developed consists of the following 5 principles, which can be called RECAP as an acronym:The “R” in RECAP stands for removing uncertainty.The “E” in RECAP stands for explaining the environment.The “C” in RECAP stands for being calm and sincere.The “A” in RECAP stands for assume nothing.The “P” in RECAP stands for personal connection.

### 2.2. Designing the NOC Course

At this point, in the quality improvement project process, the RECAP principles we wrote and the framework we developed were designed, refined, modified, and ready to be implemented with nursing and PCA staff. There is general agreement that didactic learning should be supplemented with case-based scenarios or simulation-based teachings to allow learners to apply the concepts previously taught through didactic methodologies [14,15,16]. There is less agreement on what type of case-based learning is best: standardized patients, role play, or drafted scenarios for group discussion. Even in controlled cluster-randomized trials, there was little appreciable difference in nursing staff knowledge, skills, or self-efficacy when comparing role play-based learning and standardized patients [17].

Recognizing the mixed study results from the literature, as well as recognizing the impracticality of using standardized patients to conduct NOC education with thousands of staff across the healthcare system, we opted to create case-based learning that would not rely on standardized patients. We also recognized that roleplay is time-intensive, typically requiring one-on-one or small group sessions, making it impractical to implement with system-wide staff within a six-month timeframe.

Nonetheless, we committed to the best practice of case-based learning, allowing nursing and PCA staff to apply the concepts previously taught through didactic methodologies. Therefore, we created a three-part learning. For the first component of learning, we developed an NOC video to contextualize why NOC is crucial to patient and family experience and key to promoting safety. Nursing and PCA staff watched this 10 min video module, involving one of our nurses explaining how she felt as a patient in our hospital and how she used those learnings to inspire her to approach NOC differently as a nurse.

For the second part of the learning, we conducted 1 h, live, in-person instructor-led courses where nursing and PCA staff learned the NOC principles (the RECAP model discussed above), and then they applied those learnings through rich reviews and small- and large-group discussions of case studies we wrote. The cases were written by our NOC Committee using real-life scenarios they experienced as nursing and PCA staff, coupled with patient feedback we received through the HCAHPS survey or informally through nurse recollections.

An example of one case scenario that we wrote was using a “fake” PCA named Ben. Ben tells his patient, Mr. Gibson, that he needs to “check vitals and turn him”. When Mr. Gibson asks why, Ben responds, “It’s one of the things we have to do every two hours…” As course facilitators, we asked the PCA participants in the class to elaborate on why this NOC was insufficient and what could have been said to make the ideal experience using RECAP principles.

Then, facilitators continued with the same encounter to add complexity to the case by having Ben pull the pillow from behind Mr. Gibson, saying, “Mr. Gibson, I’m going to remove this pillow. You may feel like you’re falling as I grab the turn sheet and pull you my way. Don’t worry, you aren’t going to fall out of bed. I’ve got you and I’m going to keep you safe”. At this juncture, facilitators asked the PCA participants to describe which RECAP principles were used and how they were used. Facilitators asked participants to describe how the patient would feel as a result of this explanation.

Facilitators continued with the case. As Ben turns Mr. Gibson, Mr. Gibson says, “Wait! Hold on! I’m not ready!” Yet, Ben continues to turn him, moving him from his left side to his right side. Ben says, “It’s OK. We’re already done”, and puts the pillow behind Mr. Gibson. Annoyed, Mr. Gibson states, “This hurts my old shoulder injury. I was more comfortable on my other side”. Ben states, “Our policy says you have to change sides every two hours. I’ll be back around 10 to turn you back.”

To close this case, facilitators asked PCA participants how this NOC was insufficient and what could have been executed better. To deepen the learning, facilitators asked participants to reflect on how patients feel when healthcare professionals reference policy or practice without further elaboration. Facilitators asked participants why it is important to confirm the patient is ready before we perform any physical activity, with facilitators prompting learners that, in the absence of confirming the patient is ready for physical adjustments, they could feel like we are conducting something *to* them rather than *for* them. The facilitator would then ask the group to explain the importance of assessing patient comfort levels after we perform any physical activity.

Other cases provided during this second component of the NOC course were directed at eliciting pain needs and narrating pain management plans, narrating medication purposes and side effects, and how to narrate when there are language barriers on the part of the healthcare professional or the patient. All case studies were written in a way to prompt learner small-group and large-group discussions.

The third component of the course was validations and coaching, which is similar to auditing, but we prefer the term “validating” because it connotes a collaborative, two-way process between the nurse or PCA learner and the person conducting the observations, whereas “observing” or “auditing” can feel as though the learner is “on stage” or being graded. Specifically, after the learners watched the 10 min video and attended and participated in the 1 h case-based discussion course, validators (who were all trained by members of the NOC Committee) observed and coached nursing and PCA staff in real-time on the floor. Using a template we designed to assess the quality of the narration (Figure A1 in Appendix A), validators observed the nursing staff in approximately three different patient interactions involving NOC, and validators provided them with constructive feedback after observations.

The coaching was designed to be collaborative and interactive—an experience where the learner described where or how she or he struggled in narration and what specifically she or he would like feedback on. To that end, we termed this component of the learning as “validations”, to suggest constructive, affirming teaching, rather than “auditing”, recognizing that most learners found “auditing” to be associated with personal failure or personal critique.

To reinforce the constructive–collaborative teaching model, the validator provided several affirming, positive behaviors they identified during the staff observations, and the validators were encouraged to identify up to three opportunities to maintain a positive collaborative coaching experience. Finally, validators were trained to avoid checklisting, because the NOC Committee contends that conforming to a checklist can unintentionally promote the very task-based communication behaviors our patients described as feeling rote and transactional.

## 3. Results

### Assessing the Outcomes

The next component of the PDSA quality improvement process requires studying the impacts of the project. The initial systemwide launch across seven hospitals occurred within a 6-month timeframe between February 2023 and July 2024. There was a total of 276 course offerings conducted by 30 facilitators, and 7341 nurses and PCAs completed the course. The evaluations reflected that 99% of learners believed their learning was improved by the course, with comments such as these: “I believe the Narration of Care (NOC) course has had a significant impact on our teams. By keeping this valuable information readily accessible to staff through NOC courses and reinforcing it through consistent hardwiring by our nursing leaders, we’ve seen sustained success in the nursing communication domain. While leading these classes, I frequently heard team members express that they learned something new and were excited to incorporate it into their daily practice. It’s inspiring to witness their enthusiasm for enhancing their communication with patients”.

Approximately 30% of staff were validated within that same timeframe, with validators finding that staff excellently maintained a calm, sincere attitude, and easily developed and maintained personal connections. The validators’ main areas of opportunity for staff highlighted the importance of removing uncertainty and explaining the environment. Specifically, these were the main areas of opportunity identified by validators:Narrate the side effects of medications;Narrate why you are turning off alarms, and whether the readings are normal or not;Narrate intravenous flushing with saline and potential discomforts it may cause;Narrate pain management plans—what you will do to address pain needs, within what timeframes, how you will reassess pain needs, and when you will escalate.

We describe the pre- and post-intervention impacts to HCAHPS scores in detail in a companion article, which should be printed in this same journal. As a system, there was a 3-point improvement of scores in relevant nursing communication domains.

In the companion article, we also detail the feedback from learners, both immediately post-course, as well as re-assessments six months later. The companion article provides a more detailed comment thematic analysis of participants’ and validators’ feedback, as well as scores across all HCAHPS questions and all nurse evaluations.

## 4. Discussion

Our aim was to develop a framework for teaching nurses and PCAs on how to effectively narrate care. We noticed a gap in the literature providing practical guidance and felt that nurses and PCAs would be left to develop narration skills on their own, learning by trial and error, and, in doing so, perhaps failing to meet patients’ needs, which was supported by our HCAHPS data and discussions with patients.

The preliminary results of the implemented RECAP framework, which will be discussed more elsewhere, suggests that nurses and PCAs felt the course teaching them how to use the RECAP framework advanced their skills and provided many practical strategies for where, how, and when to effectively narrate care.

That said, there were several barriers and unforeseen challenges in the rollout and development of the course. For instance, the courses were very discussion-based with high degrees of audience engagement. But, with high degrees of audience engagement, there is a high degree of unpredictability in how long the course will run, how best to redirect the audience to finish the learning, and how to best facilitate discussion where the learners become “stuck” on a particular concept or case.

For instance, one of the more challenging cases involved a language barrier on the part of the nurse. The patient in our hypothetical case expresses frustration about not being able to understand the nurse’s discharge education due to her accent. When we asked learners what the nurse could do to narrate differently or more effectively to meet the patient’s learning needs, several learners believed that by expressing frustration and dissatisfaction, the patient was acting inappropriately. What the facilitators intended to elicit—which required significant prompting—is that, when the patient has the language barrier, we often provide assistive resources, like interpreters or translation technology services, but, when the healthcare professional is the one that exhibits a language barrier, we should be equally language-aware and resourceful to maintain equity. The nurse, in our example, could have acknowledged her accent in talking with the patient, made the patient feel comfortable in asking questions, and used teach-back methods to elicit patient understanding. As other researchers have noted, cultural competency requires deep inner-reflection and awareness [18,19], which our facilitators found challenging to cultivate in one case, suggesting a need for deeper, more extensive learning over time.

An additional challenge was that, during some course sessions, facilitators detected PCA and nursing differences in opinion over whose responsibility it would be to narrate when there are pain needs. For instance, several PCAs felt that, when a patient expresses pain, they should be able to defer that entirely to the nurse. Our facilitators, on the other hand, taught PCAs that they too should narrate—by acknowledging and sympathizing with the expression of pain, quickly addressing it in real-time by calling the nurse inside of the room, and describing how and when she as the PCA would make sure that pain needs were fully addressed.

Unfortunately, there has been limited research on the impact of nurse–PCA relationships on patient safety and clinical outcomes [20]. Systematic reviews suggest a need for heightened delegation practices and clear role delineation between nursing and PCAs and linking such interventions to patient outcomes, a recommendation we would support given what could be perceived as a lack of clear role delineation between nurses and PCAs on patient expressions of pain needs.

Despite these and other challenges, the RECAP principles and framework were taught and received well by learners. There are implications for nursing and PCA practice in using and maintaining the RECAP framework. If organizations implement and adopt a version of our work, nurses and PCAs can comfortably and confidently enter the profession knowing the purpose or narrating care, its many benefits, and how to practically go about sufficiently narrating, and what would constitute insufficient narration. Newly graduated nurses want to be accepted into the team, and some research suggests that their own uncertainty and self-doubt undermines their ability to feel included [21]. By providing more certainty and confidence in narration skills, perhaps new graduates can feel included earlier in their work transition.

If the RECAP framework and course are implemented for new hires during the orientation process, hospitals, in turn, can specify and clearly articulate their expectations for nursing and PCA staff’s narration with patients, what would make for a strong, compassionate process, and what would be inadequate from the perspective of nurse leaders. By covering the course during orientation, there would be a culture of excellence standard set throughout the organization and hopefully maintained through refreshers on a routine basis.

For more experienced nurses, they can use the RECAP framework to reflect on their own practices and perhaps strengthen or refreshen existing skills. Previous research has demonstrated that experienced nurses report being unaware of self-directed learning and believe that educational activities are not well tailored to meet their needs [21]. However, they are most receptive to courses that highlight patient-specific outcomes and patient needs. Narration of care is patient-focused and can positively impact clinical outcomes [21]. By providing refresher courses with how narration of care influenced patient outcomes in a positive or negative way, perhaps experienced nurses would be inclined to take refresher courses to strengthen or refresh existing skills.

By developing a structured approach, using the RECAP principles, and developing and implementing a course that allows nurses and PCAs to apply the RECAP principles to real-life examples, our hope is to cultivate a sense of confidence and competence in nurses’ and PCAs’ narration skills across a range of specialties and years of practice or experience.

## 5. Conclusions and Future Research

The working phase in the nurse–patient relationship is likely the most substantive component of the nurse–patient relationship, and yet it is the phase that is most underdeveloped. While the behavioral aspects (what the nurse carries out) during the working phase has received attention, little has been written on the communication aspect (what the nurse says) during the working phase to promote and maintain a healthy, therapeutic relationship. Instead, there are oblique references to “patient-centered care” and “communication”, without delineating what and how to say it.

We contend that, without practical guidance on how to narrate care, nursing and PCA staff are prone to narrating in a way that likely resembles performative, task-driven, and rote-like communication behaviors that have already been identified in the literature as prevalent in healthcare [12,17] and also supported by our internal data and patient discussions. When this style of communication occurs, the nurse–patient relationship becomes fractured or undermined [22,23].

In an effort to promote healthy communication practices and provide practical guidance, we developed a framework that can be implemented in any organizational setting. In our appendix, we provide course and validation materials that can be implemented seamlessly, in the hope that other organizations and the nursing practice at large can benefit from this work. Future work should focus on sustainment and iterative empirical investigation over time—whether and how an NOC course can be sustained and whether and how “refresher” mini-courses with different case vignettes can be maintained.

## Data Availability

The data presented in this study are available on request from the corresponding author due to the fact that the data will be published in a forthcoming, companion article.

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
