# Peer review of "Developing and Implementing a Narration of Care Framework to Teach Nurses When and How to Narrate Care"

_nursrep, 2025, doi:10.3390/nursrep15070244_

Round 1

Reviewer 1 Report

Comments and Suggestions for Authors

In this report narrative care is considered as an implementation in standard nursing for the improvement of healthcare. The authors assessed patients with interviews to get feedback on the importance of this approach in everyday practice. The narratives applied were for the side effects of medications, alarms and their readings, intravenous saline, and pain management. The principles of narrative design are based to the RECAP principles: (1) removing uncertainty, (2) explaining the environment, (3) calm and sincere, (4) assume nothing, (5) personal connection. Overall, this patient-oriented approach of nursing provides important improvements of healthcare.

Author Response

Thank you for your comments and your thoughtful review. We do not see any requested changes at this time. Thank you. 

Reviewer 2 Report

Comments and Suggestions for Authors

the aim and objectives are not clearly indicated, one has to search within the document, however, the methodology, results and conclusion are well aligned, consistent and contribute to new knowledge in nursing practice.

Author Response

Thank you for the time you took in carefully reviewing the paper and outlining such thoughtful edits. We believe we successfully incorporated most, if not all, of your suggestions, resulting in a stronger paper. We also appreciate your kindness in providing constructive feedback. Thanks again for your time, consideration, diligence, and thought.

Reviewer 3 Report

Comments and Suggestions for Authors

Please see attached

Author Response

Please see attached comments and revisions. 

Reviewer 4 Report

Comments and Suggestions for Authors

Dear authors,

Thank you for your contribution.  I appreciate the effort and time you have devoted to carrying out a qualitative analysis of an important topic such as narrative care and proposing a conceptual framework consisting of 5 principles called RECAP in this area.

However, the study would benefit from several revisions to increase clarity.  Below are specific suggestions for improvement:

The abstract does not accurately reflect the content of the study.

The introduction discusses gaps in narrative care, but does not sufficiently address previous systematic or large-scale reviews of the topic.

Subsection 2.3 does not clarify how the selection of the 'NOC' is carried out.  Please indicate the criteria by which the working group of the "NOC" was determined and how many people it consisted of.  Also for subsection 2.4 in the context of patient selection for NOC.

In the results section, please indicate the form of data collection from February to July 2025.  Could it be a typo?

Correct the formal editing of the text: pages 4-5.

Author Response

Thank you for the time you took in carefully reviewing the paper and outlining such thoughtful edits. We believe we successfully incorporated most, if not all, of your suggestions, resulting in a stronger paper. We also appreciate your kindness in providing constructive feedback.

Round 2

Reviewer 3 Report

Comments and Suggestions for Authors

Please see the attached comments.

Author Response

Thank you for the time you took in carefully reviewing the paper and outlining such thoughtful edits. We believe we successfully incorporated most, if not all, of your suggestions, resulting in a stronger paper. We also appreciate your kindness in providing constructive feedback.

Thanks again for your time, consideration, diligence, and thought.

  1. Reviewer #3, Comment 1: Abstract: Nursing appears to be here twice as a keyword. I suggest organizing alphabetically.

Our response/Revisions for Reviewer #3, Comment 1: We’re not sure we see “nursing” listed twice. We now list alphabetically.

  1. Reviewer #3, Comment 2: Introduction: Line 96 – I would introduce PCA before abbreviating. I think you might need another line here to explain what a PCA is. You identify them further down in the narrative (line 102).

Our response/Revisions for Reviewer #3, Comment 2: Corrected.

  1. Reviewer #3, Comment 3: Figures: I am still unclear if the figures are yours and you created and used, or if you adapted from someone else? The institution heading is throwing me off and I think that it might be best that it is not included. I would suggest typing out the content in the figures in a suitable manner that the reader can see and labeling what each figure is according to the format suggested by the journal. Right now, the figures are not referenced in text, so it is unclear what I am looking at. I would make sure each figure is clearly referred to in text, or you add something that states you attached supplementary materials for your project. If a reader were to look at this, they would have no idea what they were looking at. And then I would clearly indicate if you need to credit someone for content you did not develop on your own.

Our response/Revisions for Reviewer #3, Comment 3:  We wrote the cases, as indicated by this text:  “The cases were written by our NOC Committee using real-life scenarios they experienced as nursing and PCA staff, coupled with patient feedback we received through the HCAHPS survey or informally through nurse recollections. An example of one case scenario that we wrote…”

We removed the figures for the sake of simplicity.
